# New insights in the targets of action of dimethyl fumarate in endothelial cells: effects on energetic metabolism and serine synthesis in vitro and in vivo

Mª Carmen Ocaña[1,2,12], Manuel Bernal[1,2,12], Chendong Yang[3,13], Carlos Caro [2,4,13], Alejandro Domínguez[2,13], Hieu S. Vu[3], Casimiro Cárdenas [1,5], María Luisa García-Martín[2,4,6], Ralph J. DeBerardinis [3,7,8,9], Ana R. Quesada[1,2,10], Beatriz Martínez-Poveda [1,2,11✉] & Miguel Ángel Medina [1,2,10✉]

Dimethyl fumarate is an ester from the Krebs cycle intermediate fumarate. This drug is approved and currently used for the treatment of psoriasis and multiple sclerosis, and its anti-angiogenic activity was reported some years ago. Due to the current clinical relevance of this compound and the recently manifested importance of endothelial cell metabolism on the angiogenic switch, we wanted to elucidate whether dimethyl fumarate has an effect on energetic metabolism of endothelial cells. Different experimental approximations were performed in endothelial cells, including proteomics, isotope tracing and metabolomics experimental approaches, in this work we studied the possible role of dimethyl fumarate in endothelial cell energetic metabolism. We demonstrate for the first time that dimethyl fumarate promotes glycolysis and diminishes cell respiration in endothelial cells, which could be a consequence of a down-regulation of serine and glycine synthesis through inhibition of PHGDH activity in these cells. Dimethyl fumarate alters the energetic metabolism of endothelial cells in vitro and in vivo through an unknown mechanism, which could be the cause or the consequence of its pharmacological activity. This new discovery on the targets of this compound could open a new field of study regarding the mechanism of action of dimethyl fumarate.

[1] Universidad de Málaga, Andalucía Tech, Departamento de Biología Molecular y Bioquímica, Facultad de Ciencias, E-29071 Málaga, Spain. [2] IBIMA Plataforma BIONAND (Biomedical Research Institute of Málaga and nano medicine Platform), E-29590 Málaga, Spain. [3] Children's Medical Center Research Institute, The University of Texas Southwestern Medical Center, Dallas, TX 75390, USA. [4] Biomedical Magnetic Resonance Laboratory-BMRL, Andalusian Public Foundation Progress and Health-FPS, Seville, Spain. [5] Research Support Central Services (SCAI) of the University of Málaga, Málaga, Spain. [6] Biomedical Research Networking Center in Bioengineering, Biomaterials & Nanomedicine (CIBER-BBN), 28029 Madrid, Spain. [7] Department of Pediatrics, The University of Texas Southwestern Medical Center, Dallas, TX 75390, USA. [8] Eugene McDermott Center for Human Growth and Development, The University of Texas Southwestern Medical Center, Dallas, TX 75390, USA. [9] Howard Hughes Medical Institute, The University of Texas Southwestern Medical Center, Dallas, TX 75390, USA. [10] CIBER de Enfermedades Raras (CIBERER), Madrid, Spain. [11] CIBER de Enfermedades Cardiovasculares (CIBERCV), Madrid, Spain. [12] These authors contributed equally: Mª Carmen Ocaña, Manuel Bernal. [13] These authors jointly supervised this work: Chendong Yang, Carlos Caro, Alejandro Domínguez. ✉email: bmpoveda@uma.es; medina@uma.es

The fumaric acid ester (FAE) dimethyl fumarate (DMF) is a methyl ester of fumaric acid (FA) that has been broadly studied in several models of disease, such as inflammatory diseases, dermatological lesions and cancer. In 2013, DMF was approved by the US Food and Drug Administration (FDA) and the European Medicines Agency (EMA) for the treatment of relapsing forms of multiple sclerosis (MS), marketed under the name of Tecfidera (previously called BG-12)[1]. DMF has also been used as an anti-psoriatic drug for more than 50 years, under the brand names Fumaderm and Skilarence[2,3].

Some years ago, we hypothesized that DMF anti-psoriatic effect could be related somehow to modulation of angiogenesis. Interestingly, our group characterized DMF as an anti-angiogenic compound using in vitro and in vivo models[4]. Simultaneously, DMF was demonstrated to exert its anti-angiogenic activity through inhibition of vascular endothelial growth factor receptor 2 (VEGFR2) expression, the main receptor for VEGF-A[5]. As mentioned by Jack Arbiser, based on these data, it seems safe to say that angiogenesis inhibition plays a role in the activity of DMF and further studies on DMF mechanisms of action seem warranted[6].

Endothelial cell (EC) energetic metabolism has been shown to be essential for correct function of ECs, and hence for correct angiogenesis trigger[7]. In consequence, targeting EC metabolism has been proposed as a novel strategy for the treatment of angiogenesis-dependent pathologies[8,9]. DMF is a cell permeable FAE that can be converted into fumarate inside the cell, thus feeding the tricarboxylic acid (TCA) cycle. Diverse, cell- and dose-dependent effects of DMF on global cell metabolism have been found in different cell types. For instance, DMF exerted a differential effect on the energetics metabolism of mouse embryonic fibroblasts depending on Nrf-2 expression and time incubation[10]. Other authors found lower respiration rates in human retinal epithelial cells treated with DMF[11]. Additionally, DMF was shown to inhibit glyceraldehyde 3-phosphate dehydrogenase (GAPDH), a glycolytic enzyme, thus down-regulating aerobic glycolysis in murine activated myeloid and lymphoid cells[12]. Moreover, DMF was found to induce cell metabolism dysfunction in human pancreatic cells through inhibition of mitochondrial respiration, aerobic glycolysis and folate metabolism, possibly by targeting the enzyme methylenetetrahydrofolate dehydrogenase 1 (MTHFD1)[13]. Nevertheless, as far as we are concerned, no studies have been performed regarding the possible role of DMF in EC energetic metabolism.

In this work, we explore the potential capacity of DMF to modulate microvascular EC glucose and/or glutamine metabolism in an in vitro model of microvascular ECs. We found that DMF diminishes cell respiration while it upregulates glycolysis in human dermal microvascular ECs (HMECs). Interestingly, our results show that DMF downregulates serine and glycine synthesis pathway in these cells through inhibition of phosphoglycerate dehydrogenase (PHGDH) activity. To our knowledge, the results presented herein are the first experimental evidence showing that DMF downregulates serine and glycine synthesis pathway in ECs. The observed alteration of EC metabolism exerted by DMF could open new horizons for further characterization of its mechanism of action in angiogenesis-dependent diseases.

## Methods

**Materials**. MCDB-131 cell culture medium was obtained from Gibco (Paisley, Scotland, UK). Glucose, glutamine, serine and glycine free media were from Teknova (Hollister, CA, USA) and from US Biological Life Sciences (Salem, MA, USA). Other cell culture media, penicillin and streptomycin and trypsin were purchased from BioWhittaker (Verviers, Belgium). Fetal bovine serum (FBS) was purchased from Biowest (Kansas, USA). Dialyzed FBS (dFBS) was from Gemini Bioproducts (West Sacramento, CA, USA) and from Capricorn (Ebsdorfergrund, Germany). Matrigel was purchased from Becton-Dickinson (Bedford, MA, USA). Material for Seahorse experiments were from Agilent Technologies (Santa Clara, CA, USA). 2-NBDG was supplied by Molecular Probes (Eugene, OR, USA). L-[U-$^{14}$C]-Glutamine was acquired from Perkin Elmer (Waltham, MA, USA). L-glutamine/ammonia assay kit was from Megazyme (Bray, County Wicklow, Ireland). Anti-HIF1α antibody was from BD Biosciences (San Jose, CA, USA), anti-α-tubulin antibody was from Cell Signaling Technology (Danvers, MA, USA), anti-PHGDH antibody was from GeneTex (Irvine, CA, USA) and anti-calnexin was from Enzo Life Sciences (Farmingdale, NY, USA). D-[U-$^{13}$C]-Glucose, L-[U-$^{13}$C]-glutamine, L-[U-$^{13}$C]-serine and L-[U-$^{13}$C]-glycine were purchased from Cambridge Isotope Laboratories (Tewksbury, MA, USA). PHGDH activity assay kit was from BioVision (Milpitas, CA, USA). Plastic material for cell culture was from Nunc (Roskilde, Denmark). Formaldehyde, ethanol, Harris hematoxylin solution and eosin yellowish hydroalcoholic solution were obtained from Panreac (Barcelona, Spain). Picrosirius red was from Morphisto (Frankfurt, Hessen, Germany). All other reagents not listed on this section, including DMF and glucose and glutamine free medium, were from Sigma-Aldrich (St. Louis, MO, USA).

**Cell culture**. All cell culture media, unless otherwise specified, were supplemented with glutamine (2 mM), penicillin (50 U/mL) and streptomycin (50 U/mL). Human microvascular endothelial cells (HMECs) were kindly supplied by Dr. Arjan W. Griffioen (Maastricht University, Netherlands) and maintained in MCDB-131 medium supplemented with 10% FBS, hydrocortisone (1 μg/mL) and EGF (10 ng/mL). Human umbilical vein endothelial cells (HUVECs) were isolated by a modified collagenase treatment as previously reported and maintained in 199 medium supplemented with 20% fetal bovine serum, ECGS (30 μg/mL) and heparin (100 μg/mL)[14]. Bovine aortic endothelial cells (BAECs) were isolated from bovine aortic arches as previously described and maintained in Dulbecco's modified Eagle's medium (DMEM) containing glucose (1 g/L) and supplemented with 10% FBS[15]. Primary human gingival fibroblasts (HGF) were maintained in DMEM containing glucose (4.5 g/L) and 10% FBS. Tumor cells used in this paper (human breast carcinoma MDA-MB-231 and MCF7, and human cervix adenocarcinoma HeLa) were purchased from the ATCC (Rockville, MD, USA) and maintained in RPMI-1640, DMEM containing glucose (4.5 g/L) and EMEM, respectively, all of them supplemented with 10% FBS. All cell lines were maintained at 37° C under a humidified 5% $CO_2$ atmosphere.

**Tube formation on Matrigel by endothelial cells**. Each well of a 96-well plate was coated with Matrigel (50 μL of about 10.5 mg/mL) at 4 °C and polymerized at 37 °C for a minimum of 30 min. $7 \times 10^4$ cells were seeded in 200 μL of medium without serum. 25, 50 and 100 μM DMF were added to the wells and incubated at 37 °C. After 5 h incubation, cultures were observed and photographed with a microscope camera Nikon DS-Ri2 coupled to a Nikon Eclipse Ti microscope from Nikon (Tokyo, Japan). Closed "tubular" structures were counted using ImageJ software.

**Extracellular flux analyzer experiments**. HMECs were cultured at a density of $3 \times 10^4$ cells/well in 24-well Seahorse XFe24 plates (Agilent) and incubated at 37° C under a humidified 5% $CO_2$ atmosphere overnight in the presence or absence of DMF. Cells were washed twice with XF base medium (Agilent) and incubated with XF medium containing or not DMF and supplemented with

10 mM glucose and 4 mM glutamine or just with 4 mM glutamine at 37° C without $CO_2$ for one hour. Three initial measurements were made using XFe24 Seahorse analyzer. Then, 1 μM oligomycin, 0.6 μM carbonyl cyanide-4-(trifluoromethoxy)phenylhydrazone (FCCP), and 1 μM antimycin A and 1 μM rotenone for wells with both glucose and glutamine, or 10 mM glucose, 1 μM oligomycin and 25 mM 2-deoxyglucose (2-DG) for wells with only glutamine, were injected sequentially to each well with three measurements between each injection. Data were analyzed using Wave software and normalized to protein amount.

**Measurement of glucose, glutamine, lactate, glutamate and ammonia**. For glucose uptake after short incubation time, cells cultured in 96-well plates were treated with DMF overnight. Then, cells were washed twice with PBS supplemented with calcium and magnesium (DPBS), and then starved for 30 min with this DPBS containing or not DMF. Cells were incubated for additional 30 min with DPBS containing or not DMF and supplemented with 5 mM glucose, 0.5 mM glutamine and 100 μM 2-NBDG (2-(N-(7-Nitrobenz-2-oxa-1,3-diazol-4-yl)Amino)-2-Deoxyglucose). Relative glucose uptake was determined using a FACS VERSE[TM] cytometer from BD Biosciences (San Jose, CA, USA)[16]. Data were analyzed with BD FACSuite software. For longer incubation times, concentrations of extracellular glucose, lactate, glutamine, and glutamate of control and DMF-treated cells were determined from aliquots of media using an automated electrochemical analyzer (BioProfile Basic-4 analyzer; NOVA). Ammonia secretion was measured from a fraction of the same aliquots with a spectrophotometric assay (Megazyme) using a FLUOstar Omega microplate reader from BMG LABTECH (Ortenberg, Germany). Data were normalized to cell number.

**Measurement of glutamine oxidation**. Cells cultured in 24-well plates and treated or not with 100 μM DMF overnight were washed twice with PBS supplemented with DPBS, and then starved for 30 min with this DPBS containing or not DMF. Cells were incubated for additional 30 min with DPBS containing or not DMF and supplemented with 25 mM HEPES, 0.5 mM glutamine, 5 mM glucose and 0.5 μCi/mL L-[U-$^{14}$C]-glutamine. Media and cells were collected in round-bottom glass tubes with screw-caps. Each glass tube contained a Whatman[TM] paperfold imbibed with benzethonium hydroxide (hyamine). 400 μL 10% (v/v) $HClO_4$ were added to the closed glass tubes through the cap. Tubes were incubated for additional 30 min at 37 °C with agitation. Whatman[TM] paperfolds with $^{14}CO_2$ captured in them were mixed with scintillation liquid. A Beckman Coulter LS6500 liquid scintillation counter (Fullerton, CA, USA) was used for the measurements. All assays were performed at the Radioactive Facilities of the University of Málaga, authorized with reference IR/MA-13/80 (IRA-0940) for the use of non-encapsulated radionuclides.

**Proteomics analysis**. Control and DMF treated cells were extensively washed and frozen for their analysis. Cells were lysed using RIPA buffer on ice for 5 min and scratched. Cell extracts were sonicated and centrifuged and supernatants were collected. For protein precipitation, a modified trichloroacetic acid–acetone precipitation method (Clean-Up Kit; GE Healthcare, Munich, Germany) was used. The resultant precipitate was dissolved in bidistilled water, sonicated and centrifuged and supernatants were collected in a clean tube. Then we carried out a gel-assisted proteolysis entrapping the protein solution in a polyacrylamide gel matrix. Peptides were extracted from the gel pieces with ACN/0.1% formic acid and the samples were dried in a SpeedVac[TM] vacuum concentrator. The dried peptides from each sample were

reconstituted in 0.1% formic acid and quantified in a NanoDrop[TM] (Thermo Scientific) to equalize all samples at an identical protein concentration before analysis on the liquid chromatography-tandem mass spectrometry (LC-MS/MS) system. Mass spectrometry (MS) analysis was performed using an Easy nLC 1200 UHPLC system coupled to a hybrid linear trap quadrupole Orbitrap Q-Exactive HF-X mass spectrometer (ThermoFisher Scientific). Software versions used for the data acquisition and operation were Tune 2.9 and Xcalibur 4.1.31.9. The acquired raw data were analyzed using Proteome Discoverer[TM] 2.2 (Thermo Fisher Scientific). Normalization was performed based on specific abundance of human β-actin protein and samples were scaled to controls average. The MS proteomics data have been deposited to the ProteomeXchange Consortium via the PRIDE partner repository with the dataset identifier PXD014489[17].

**RNA extraction and gene expression analysis**. Total RNA from control and DMF-treated HMECs was extracted using Tri Reagent[TM] (Sigma-Aldrich), and RNA was purified with the Direct-zol™ RNA MiniPrep Kit (Zymo Research) according to the manufacturer's instructions. RNA quality and amount was measured using a NanoDrop ND-1000 (Thermo Scientific). cDNA synthesis was performed using the PrimeScript™ RT reagent Kit (Takara) following purchaser's instructions. mRNA expression analysis was determined using KAPA SYBR Fast Master Mix (2×) Universal (KAPA Biosystems) in an Eco Real-Time PCR System (Illumina). The following thermal cycling profile was used: 95° C, 3 min; 40 cycles of 95° C, 10 s and 54° C, 30 s. Primer sequences were as follows: *ACTB* forward: GACGACATGGA-GAAAATCTG; *ACTB* reverse: ATGATCTGGGTCATCTTCTC; *SLC2A1* forward: ACCTCAAATTTCATTGTGGG; *SLC2A1* reverse: GAAGATGAAGAACAGAACCAG. qPCR data were normalized to *ATCB* expression.

**Western blot**. Cells were lysed with RIPA (50 mM Tris-HCl pH 7.4, 150 nM NaCl, 1% Triton X-100, 0.25% sodium deoxycholate and 1 mM EDTA) or with 2x denaturing loading buffer. Samples were heated at 95° C during 5 min and separated on 10% polyacrylamide gels. Proteins were transferred to nitrocellulose membranes and blocked with 10% (w/v) semi skimmed dried milk. Blocked membranes were incubated overnight with primary antibodies (anti-HIF-1α 1:500, anti-PHGDH 1:1000, anti-α-tubulin 1:10000, anti-calnexin 1:1000), washed and incubated with the peroxidase-linked secondary anti-rabbit or anti-mouse antibody for 1 h at room temperature. Membranes were washed and finally incubated with the Supersignal® West Pico chemiluminescent substrate system (Thermo Scientific). Image captions were taken with the ChemiDoc[TM] XRS+ System (Bio-Rad) using Image Lab software or either films were revealed using a Medical film processor from Konica Minolta (Tokyo, Japan). Densitometry analyses were made with ImageJ software.

**Metabolomics and labeling experiments using stable isotopes**. Cells grown in 6-cm dishes to 80-90% confluence were washed and 10 mM glucose or [U-$^{13}$C]-glucose and 4 mM glutamine or [U-$^{13}$C]-glutamine were added for the indicated times in DMEM supplemented with 10% dFBS in the presence or absence of DMF. For experiments of serine and glycine withdrawal or labeling, 10 mM glucose, 4 mM glutamine, 0.4 mM serine or [U-$^{13}$C]-serine and/or 0.4 mM glycine or [U-$^{13}$C]-glycine were added for 24 h in serine and glycine free RPMI-1640 or DMEM supplemented with 10% dFBS containing or not DMF. For analysis of intracellular metabolites by gas chromatography/mass spectrometry (GC/MS), labeled cells were rinsed in ice-cold saline

solution and lysed with three freeze-thaw cycles in cold 80% methanol. Debris was discarded by centrifugation and 50 nmol of sodium 2-oxobutyrate were added as internal standard to the supernatants. Samples were evaporated, derivatized with N-(Tert-Butyldimethylsilyl)-N-Methyltrifluoroacetamide (MTBSTFA) and analyzed on an Agilent 7890 gas chromatograph coupled to an Agilent 5975 mass selective detector as previously described[18]. Data were acquired using MSD ChemStation software (Agilent). For analysis of intracellular metabolites by liquid chromatography/mass spectrometry (LC/MS), samples obtained in the same way but without addition of sodium 2-oxobutyrate were evaporated. Dried samples were reconstituted in 0.1% formic acid in water and 5 μL were injected into a 1290 UHPLC liquid chromatography (LC) system interfaced to a high-resolution mass spectrometry (HRMS) 6550 iFunnel Q-TOF mass spectrometer (MS) (Agilent). The MS was operated in both positive and negative (ESI+ and ESI-) modes. Analytes were separated on an Acquity UPLC® HSS T3 column (1.8 μm, 2.1 ×150 mm, Waters, MA). The column was kept at room temperature. Mobile phase A composition was 0.1% formic acid in water and mobile phase B composition was 0.1% formic acid in 100% acetonitrile. The LC gradient was 0 min: 1% B; 5 min: 5% B; 15 min: 99%; 23 min: 99%; 24 min: 1%; 25 min: 1%. The flow rate was 250 μL/min. Data were acquired using Profinder B.08.00 SP3 software (Agilent). Intracellular relative abundance of metabolites was normalized to cell number for GC/MS samples or by total ion current (TIC) normalization for LC/MS samples and represented in a heatmap using Heatmapper[19].

**Determination of PHGDH activity in HMECs.** PHGDH activity was measured in control and DMF treated cells using a spectrophotometric assay as indicated by the supplier (Phosphoglycerate Dehydrogenase (PHGDH) Activity Assay Kit (Colorimetric), BioVision). 100 μg protein of cell lysates were added to each well along with the reaction mix and PHGDH activity was monitored at 450 nm at 37° C using a FLUOstar Omega microplate reader from BMG LABTECH (Ortenberg, Germany). Additionally, cell lysates from the control condition were added along with DMSO or 100 μM DMF to the reaction mix.

**Zebrafish caudal fin regeneration assay.** Adult wild-type zebrafish (*Danio rerio*) were maintained in water at 27 °C and an electric conductivity of 800 μS. Animals were anesthetized with 0.2 mg/mL tricaine for partial amputation of the caudal fin. At 6 dpa (days post amputation), fish were incubated in fish water containing DMSO as the vehicle control or two different doses of DMF (25 and 50 μM) for additional 24 h. After incubation, the regeneration border of the fins was cut, immediately weighted and frozen in liquid nitrogen and conserved at –80 °C prior [1]H HR-MAS NMR analysis. Alternatively, the regeneration border was used for histological preparation. Non-regenerated fins were used as basal situation.

**Histology of caudal fins from zebrafish.** Tissues from sections of zebrafish fins were fixed for 48 h in Dietrich's Fixative solution composed by 40% formaldehyde (pH 4 buffered), 99.5% acetic acid glacial, 96% ethanol and distilled water. The solution was changed every 24 h. Then, samples were dehydrated through graded ethanol, and embedded in paraffin for 2 h under stirring-vacuum at 56° C. All samples were sectioned at 7 μm thickness, deparaffinized, rehydrated, stained, dehydrated, cleared in xylene and mounted on commercial glass slides for visualization in an Olympus Slideview VS200 digital slide scanner with indirect polarized light (210°).

The staining protocols were hematoxylin and eosin (H&E, Harris hematoxylin solution and 1% eosin yellowish hydroalcoholic solution) and Picrosirius red (Picrosirius red and Harris hematoxylin solution).

**Sample preparation and [1]H HR-MAS NMR spectra acquisition and processing.** Sections from zebrafish fins for the same experimental condition were pooled and introduced into a High-Resolution Magic Angle Spinning (HR-MAS) insert, and weighed on a high-precision scale. The insert was then filled up with a standard solution containing PBS, 1 mM TSP-d4 (3-(trimethylsilyl)2,2,3,3-tetradeuteropropionic acid sodium salt) and $D_2O$. Finally, the insert was introduced into a ZrO2 rotor and placed in the magnet pre-cooled to 4 °C.

Spectra were acquired on a Bruker Avance™ 600 MHz spectrometer, equipped with an Avance III console and a 4 mm TXI HR-MAS probe with a z-gradient aligned along the magic angle (Bruker BioSpin, Ettlingen, Germany). Water-suppressed 1D [1]H NMR spectra were acquired using a Carr-Purcell-Meiboom-Gill (CPMG) sequence with the following acquisition parameters: 12 kHz spectral width, 64k data points, 64 scans, 1 ms echo time (2τ) with a total echo time of 130 ms, and 5 s relaxation delay. Water presaturation was applied during the relaxation delay. Then, spectra were Fourier transformed with a 0.5 Hz exponential filtering, the baseline corrected, and an artificial reference signal, ERETIC (Electronic Reference To access In-vivo Concentrations), was added using the ERETIC 2 method implemented in TopSpin 3.5.pl7. Metabolite identification and quantification were performed with the software Chenomx (v8.3, Chenomx Inc., Edmonton, Ca), using the ERETIC signal as the concentration reference.

**Animals and ethical statement.** Animal experiments were performed in accordance with the Spanish and European Guidelines for Care and Use of Laboratory Animals (R.D. 53/2013 and 2010/62/UE) and approved by the Animal Ethics Committee of the University of Málaga, and the Highest Institutional Ethical Committee (Andalusian Government, accreditation number 04/05/2021/069).

**Statistics and reproducibility.** Results from [1]H HR-MAS NMR are expressed as the data obtained for the sum of all the fin sections pooled for each experimental condition. All other results are expressed as means ± SD. Data shown for extracellular flux analysis are for a representative experiment with 3 replicates, which was repeated three times. Metabolomics and labeling experiments were performed once with 3 replicates for GCMS data and 4 replicates for LCMS data. Data in the remaining figure panels reflect three independent experiments unless otherwise specified. For proteomics analysis, abundance ratio p-values were calculated by ANOVA based on background population of peptides and proteins, and values of $p < 0.01$ were considered to be statistically significant. For the rest of the experiments, statistical significance was determined using the unpaired, two-sided Student $t$ test and values of $p < 0.05$ were considered to be statistically significant. In all figures, the $p$ values were shown as: $*p < 0.05$; $**p < 0.01$; $***p < 0.001$; $****p < 0.0001$.

**Reporting summary.** Further information on research design is available in the Nature Portfolio Reporting Summary linked to this article.

## Results

**DMF inhibits capillary tube formation in microvascular endothelial cells.** Antiangogenic activity of DMF has been

previously described in an in vitro model of macrovascular ECs[4], but its effect on microvascular ECs has not been assessed before. In order to determine the concentrations of DMF that interfere with angiogenesis in vitro in HMECs (microvascular ECs), we evaluated the dose of DMF able to totally inhibit tube formation in these cells. For this aim, we performed a capillary tube formation assay on Matrigel using increasing DMF concentrations. Consistent with the previously published effect on macrovascular ECs, we confirmed the total inhibition of tubular-like morphogenesis on Matrigel by 50 and 100 μM DMF in microvascular ECs (Supplementary Fig. S1). Then we decided to use 50 μM for further experiments.

### DMF diminishes respiration while favors glycolysis in HMECs.
As a first approximation to test the capacity of DMF to modulate global energetic metabolism in HMECs, oxygen consumption rate (OCR) and extracellular acidification rate (ECAR), as estimators of oxidative phosphorylation (OXPHOS) and glycolysis, respectively, were measured using a Seahorse flux analyzer. Obtained data showed that cells incubated for 20 h with 50 μM DMF had a

higher glycolytic rate than control cells ($p < 0.001$) (Fig. 1A, B and Supplementary Fig. S2A–D).

Interestingly, this increased glycolytic rate observed in DMF-treated HMECs was correlated with an increased glucose uptake in these cells. HMECs cultured overnight with several concentrations of DMF were exposed to 30 min fasting in DPBS in the presence of DMF, followed by additional 30 min incubation in the presence of glucose and glutamine. Glucose taken up during those 30 min in the presence of DMF was measured, revealing that DMF increased glucose uptake in HMECs in a dose-dependent manner ($p < 0.05$) (Fig. 1C). Since 100 μM exerted a greater effect than 50 μM without compromising cell viability, we decided to keep using 100 μM DMF for further experiments.

In order to elucidate if the observed effect of DMF on glucose metabolism could be dependent on transcriptional regulation, we compared results of glucose uptake after overnight treatment with DMF with shorter incubation time (DMF added during the last 30 min incubation with glucose). Our data showed that 100 μM DMF incubated in short time (30 min before measurements) increased glucose uptake in HMECs to a lesser extent than after overnight incubation, yet the increase was statistically significant

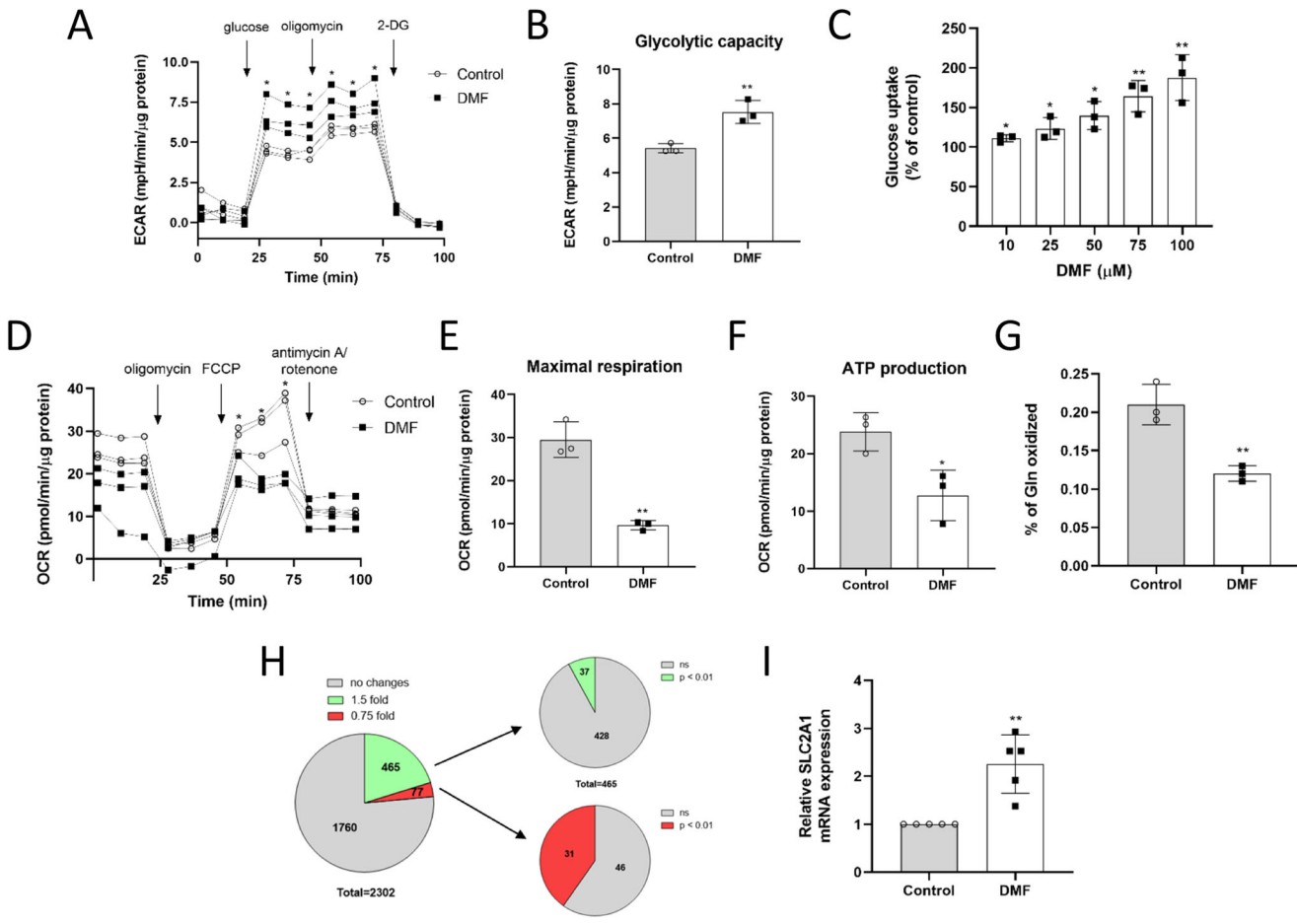

**Fig. 1 DMF increases glycolysis and diminishes OXPHOS in HMECs. A** ECAR was measured in cells treated with 50 μM DMF and (**B**) glycolytic capacity rate was calculated. **C** Glucose uptake after 30 min incubation with 5 mM glucose and 0.5 mM glutamine in cells treated with different doses of DMF for 16 h. **D** OCR was measured in cells treated with 50 μM DMF and (**E**) maximal respiration and (**F**) ATP production were calculated. **G** Glutamine oxidation after 30 min incubation with 5 mM glucose, 0.5 mM glutamine and 0.5 μCi/mL L-[U-14C]-glutamine after treatment with 100 μM DMF for 16 h. **H** Proteomics analysis of cells treated with 100 μM DMF for 24 h. **I** *SLC2A1* mRNA expression in cells treated with 100 μM DMF for 24 h. Data are expressed as means ± SD. Representative figures from three independent experiments with three replicates are shown in (**A**, **B**) and (**D**–**F**). Means and SD for three independent experiments are shown in (**C**, **G**). Means and SD for five independent experiments are shown in (**I**). Statistical significance was determined using the unpaired, two-sided Student *t* test and the resulting *p* values are shown in the figure as follows: *$p < 0.05$; **$p < 0.01$; ***$p < 0.001$ versus untreated control. An ANOVA was performed for calculating abundance ratio *p* values of (**H**).

in both conditions ($p < 0.05$) ($116.2 \pm 4.2\%$ and $187.6 \pm 29.1\%$ data of fold compared to the control condition in short time and overnight DMF treatments, respectively). These results suggest the implication of transcriptional regulation in the DMF-enhanced glucose metabolism in ECs, although additional shorter-term mechanisms may also contribute to this effect.

Additionally, we determined that OCR was lower in DMF-treated cells (Fig. 1D–F and Supplementary Fig. S3E–J). Since intracellular glutamine is mainly incorporated into the TCA and oxidized, we tested glutamine oxidation in HMECs in presence of 100 µM DMF after overnight incubation, using the same experimental conditions described in glucose uptake assays. As shown in Fig. 1G, 100 µM DMF halved glutamine oxidation in HMECs ($p < 0.01$), indicating an effect in the metabolic use of this amino acid.

Previously described experiments were performed in nutrient-limited conditions, assuring the detection of direct effects of DMF on the metabolic substrates of interest. On the other hand, in order to assess glucose and glutamine uptake, as well as lactate, glutamate and ammonia secretion in a more complex mixture of nutrients, we cultured cells overnight or for 24 h incubation in the presence of 100 µM DMF in complete medium. Either way, glucose uptake and lactate secretion were increased in HMECs treated with DMF ($p < 0.01$) (Supplementary Fig. S3A–C). On the contrary, DMF reduced glutamine uptake in HMECs ($p < 0.05$), whereas glutamate release to the medium was slightly higher in treated cells (Supplementary Fig. S3C, D). Regarding ammonia production, none or small differences were found in the presence of DMF (Fig. S3C, D). All these results point to a DMF-mediated upregulation of glycolysis in HMECs, whereas oxidative metabolism seems compromised in presence of this compound. Obtained results were similar in both timepoints, and hence 24 h incubation was preferred for next experiments.

**DMF has differential effects in different cell lines**. In order to see whether the observed effects of DMF were specific to HMECs, we also tested glucose uptake and glutamine oxidation in different cell lines treated with DMF, including macro- and mesovascular ECs (BAECs and HUVECs), two breast adenocarcinoma cell lines (MDA-MB-231 and MCF7), a cervix adenocarcinoma cell line (HeLa) and fibroblasts (HGF), in order to cover non-microvascular ECs, different tumor cell lines and a non-transformed cell line different from endothelium.

As shown in Supplementary Fig. S4A, the effect of 100 µM DMF overnight incubation on glucose uptake was also found in all the tested cell lines ($p < 0.05$), suggesting that DMF could be targeting a common mechanism in all of them. Regarding glutamine oxidation, a slight inhibitory effect, yet not statistically significant, was found in HUVECs after DMF treatment, whereas this DMF-induced reduction was significant in tumor MDA-MB-231 cells ($p < 0.05$) and no effect was found in HeLa cells (Supplementary Fig. S4A).

**DMF upregulates GLUT1 expression**. Due to the greater effect of DMF on glucose uptake after longer incubation, we hypothesized that this compound might modulate glucose and/or glutamine metabolism through modulation of gene and/or protein expression. To test this premise, a quantitative proteomics analysis was performed in samples from HMECs treated with 100 µM DMF for 24 h. We considered an upregulation on protein expression when at least a 1.5-fold increase in the abundance ratio (DMF/DMSO) was found and a downregulation of those proteins with a 0.75-fold or lower expression in the abundance ratio (DMF/DMSO). A total of 2302 proteins were identified with a high confidence level and at least two peptides detected. Of

those, 465 presented a ≥ 1.5-fold increase and 77 a ≤ 0.75-fold expression. However, we only considered statistically significant those changes with a p-value lower than 0.01, thus selecting 37 upregulated proteins and 31 downregulated after DMF treatment (Fig. 1H and Supplementary Tables S1–S3).

Among the upregulated proteins, glucose transporters GLUT1 and GLUT14, also known as solute carrier family 2, facilitated glucose transporter member 1 (*SLC2A1*) and member 14 (*SLC2A14*), respectively, expressions were found to be 3.72-fold and 4.06-fold compared to the control condition, respectively ($p < 0.01$) (Supplementary Table S2). Moreover, DMF also increased mRNA *SLC2A1* expression in these cells (Fig. 1I). Since GLUT1 is under the transcriptional control of HIF-1α, and DMF was shown to stabilize HIF-1α in human embryonic kidney cells, we checked HIF-1α protein levels in HMECs treated with DMF[20]. However, we did not detect any HIF-1α in normoxia with DMF (Supplementary Fig. S5). Thus, this increase in GLUT1 expression was not likely the consequence of a stabilization of HIF-1α in normoxia in the presence of DMF, discarding this possibility and pointing to a different mechanism of action responsible of the observed increase in glucose transporters expression induced by DMF.

**DMF affects aspartate and TCA cycle intermediates levels**. Next, we performed steady-state metabolomics in complete medium in order to study the possible changes in the intracellular pool of several metabolites as a consequence of the deregulated glycolytic and oxidative metabolism in DMF-treated cells. Among other changes, we observed that aspartate levels were drastically lower in DMF-treated HMECs ($p < 0.0001$) (Fig. 2A and Supplementary Fig. S6). Of note, aspartate is absent in DMEM formulation, which we used for these experiments, and hence cells need to synthetize it. However, we also performed this experiment in cells cultured in RPMI-1640 medium, which contains aspartate and, in these conditions, aspartate levels in DMF-treated cells were also lower, but to a lesser extent than when cells were cultured in aspartate-free medium ($73,21 \pm 3,82\%$ versus control in RPMI medium, whereas the reduction was $23,32 \pm 0,93\%$ versus control in DMEM medium). Interestingly, aspartate synthesis is regulated by the electron transport chain (ETC) activity[21]. Since DMF treatment suppressed respiration and glutamine oxidation in HMECs, we also performed stable isotope-labeling studies using glutamine labeled with carbon-13 in its five carbons ([U-13C]-glutamine) to follow the labeling of TCA intermediates (Fig. 2B). As expected, labeling of aspartate, fumarate, malate and citrate from glutamine was lower in DMF-treated cells ($p < 0.05$) (Fig. 2C–F), corroborating a lower incorporation of glutamine into the TCA cycle. The fumarate pool was not affected (Fig. 2G), probably due to external addition of fumarate from DMF. Intracellular malate levels were lower ($p < 0.0001$) (Fig. 2H). However, citrate levels were increased after DMF treatment ($p < 0.001$) (Fig. 2I), which may reflect inhibition of the TCA cycle. Supplementary Data for these experiments are available in Supplementary Data 2.

**DMF decreases serine and glycine synthesis and favors extracellular serine and glycine uptake in HMECs**. Interestingly, by means of the steady-state metabolite experiment, higher levels of intracellular glycine were found in DMF-treated cells ($p < 0.001$) (Fig. 3A and Supplementary Fig. S6). Serine levels were slightly higher, yet not statistically significant (Fig. 3B). Again, we performed stable-isotope-labeling studies, this time using glucose labeled at all six carbons with carbon-13 ([U-13C]-glucose), in order to check whether serine and glycine synthesis from glucose was boosted in the presence of DMF. The endogenous synthesis of

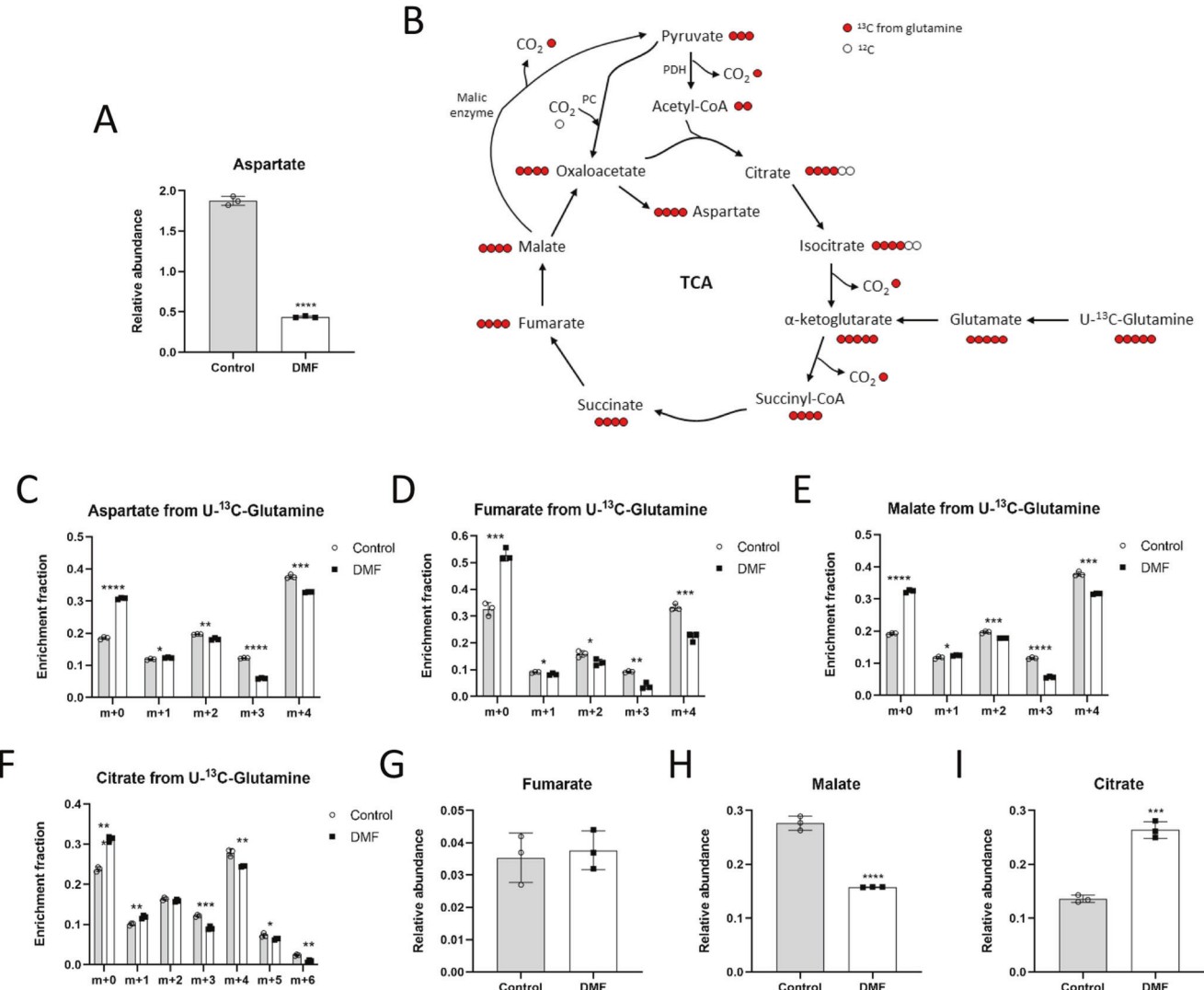

**Fig. 2 DMF diminishes incorporation of glutamine into the TCA cycle in HMECs. A** Intracellular aspartate levels in cells treated with 100 μM DMF for 24 h. **B** Scheme of TCA cycle illustrating labeling from [U-$^{13}$C]-glutamine. **C** Fractional labeling of aspartate, (**D**) fumarate, (**E**) malate and (**F**) citrate from [U-$^{13}$C]-glutamine in cells treated with 100 μM DMF for 24 h. **G** Intracellular fumarate, (**H**) malate and (**I**) citrate levels in cells treated with 100 μM DMF for 24 h. Data are expressed as means ± SD for an experiment with three replicates. Statistical significance was determined using the unpaired, two-sided Student $t$ test and the resulting $p$ values are shown in the figure as follows: $*p < 0.05$; $**p < 0.01$; $***p < 0.001$; $****p < 0.0001$ versus untreated control.

serine and glycine starts from glucose, which through glycolysis, converts after several steps into 3-phosphoglycerate (3-PG). This glycolytic intermediate is the substrate of phosphoglycerate dehydrogenase (PHGDH). The resultant 3-phosphohydroxypyruvate (PHP) is then converted into 3-phosphoserine (P-Ser) through phosphoserine aminotransferase (PSAT), and this P-Ser is finally the substrate of phosphoserine phosphatase (PSPH), resulting in the synthesis of serine. Finally, glycine is the product of the enzyme serine hydroxymethyltransferase (SHMT) from serine (Fig. 3C).

Surprisingly, we found that not only serine m + 3 from glucose was lower after 24 h incubation with 100 μM DMF ($p < 0.0001$) (Fig. 3D), but an even greater decrease in glycine m + 2 was detected ($p < 0.0001$) (Fig. 3E). No changes in 3-PG m + 3 labeling or intracellular levels were found (Fig. 3F, G), which could have been expected due to the higher glycolytic activity in DMF-treated cells. Similar effects were found in HMECs treated with 100 μM DMF overnight or with 50 μM DMF for 24 h (Supplementary Figs. S7, S8). Supplementary Data for these experiments are available in Supplementary Data 2.

Due to the presence of extracellular serine and glycine in the medium we used for metabolomics and isotope tracing analysis, we checked whether depleting serine and glycine from the medium affected metabolism of HMECs. Our data showed that serine and glycine withdrawal did not affect the observed effect of DMF on glucose uptake or lactate production in HMECs (neither level in control conditions) compared to conditions with extracellular serine and glycine (Supplementary Fig. S9).

Regarding serine and glycine synthesis from glucose, on the one hand serine m + 3 was higher in control and DMF-treated HMECs incubated in the absence of both serine and glycine ($p < 0.0001$) (Fig. 4A), indicating the need for serine synthesis when no extracellular serine is available. Serine pools were higher in DMF-treated cells when serine and glycine were present in the medium ($p < 0.001$), whereas serine levels were low in serine and glycine depleted medium in both conditions ($p < 0.001$) (Fig. 4B). Glycine m + 2 was also higher in control HMECs when no extracellular serine and glycine was available ($p < 0.01$) (Fig. 4C). However, DMF-treated cells failed to increase glycine m + 2

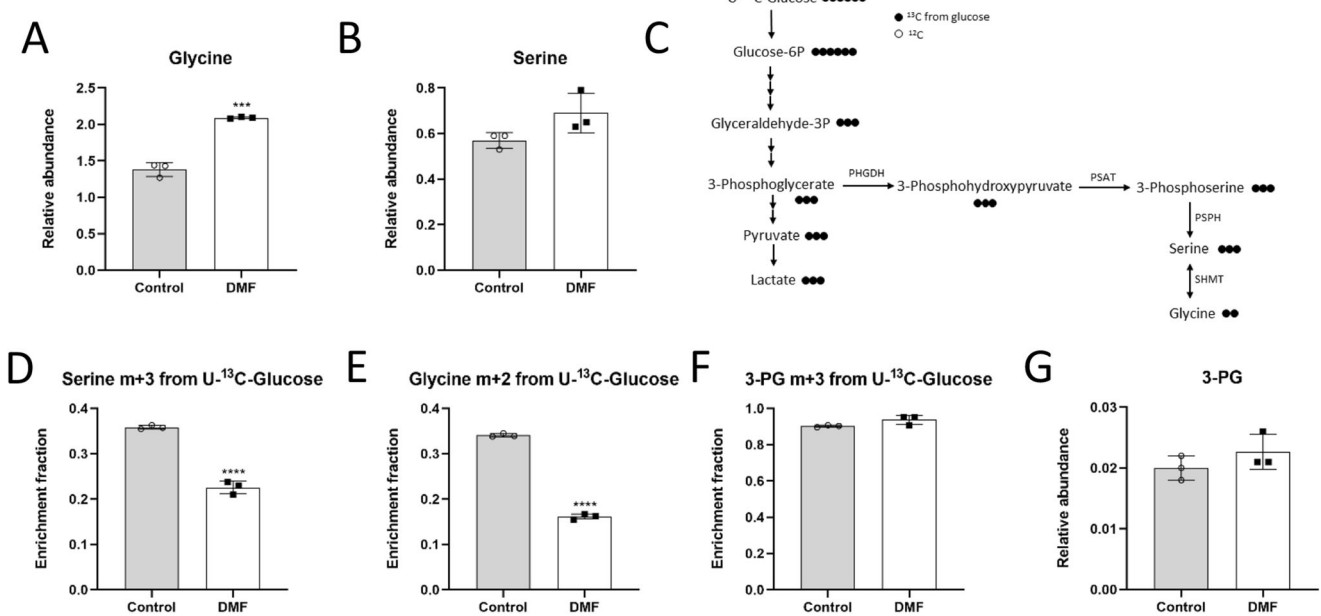

**Fig. 3 DMF decreases serine and glycine synthesis from glucose in HMECs. A** Intracellular glycine and (**B**) serine levels in cells treated with 100 μM for 24 h. **C** Scheme of glycolysis and the serine and glycine synthesis pathway illustrating labeling from [U-$^{13}$C]-glucose. **D** Fractional labeling of serine, (**E**) glycine and (**F**) 3-PG from [U-$^{13}$C]-glucose in cells treated with 100 μM DMF for 24 h. **G** Intracellular 3-PG levels in cells treated with 100 μM DMF for 24 h. Data are expressed as means ± SD for an experiment with three replicates. Statistical significance was determined using the unpaired, two-sided Student $t$ test and the resulting $p$ values are shown in the figure as follows: ***$p < 0.001$; ****$p < 0.0001$ versus untreated control.

labeling from glucose in the absence of these two amino acids to the same extent as they did with serine ($p < 0.01$ compared to cells treated with DMF in the presence of serine and glycine) (Fig. 4C). Furthermore, intracellular glycine levels were increased in DMF-treated cells when there was serine and glycine in the medium ($p < 0.001$), but the glycine pool was almost totally depleted during serine and glycine withdrawal ($p < 0.01$) (Fig. 4D). No remarkable changes were found in either 3-PG labeling or intracellular pool between control and DMF-treated cells in conditions with or without extracellular serine and glycine (Fig. 4E, F). Together, these results suggest that DMF-treated ECs had their de novo synthesis pathway compromised, while the intracellular pool of these two amino acids was higher compared to control cells. Supplementary Data for these experiments are available in Supplementary Data 3.

Therefore, we next checked if an increase in extracellular serine and glycine uptake was taking place in DMF-treated cells. For that aim, we used serine and glycine labeled with carbon-13 in all their carbons ([U-$^{13}$C]-serine and [U-$^{13}$C]-glycine). In order to avoid interferences, extracellular glycine was absent in medium supplemented with labeled serine, whereas serine was not added to the medium with labeled glycine, since these two amino acids can be converted into each other through SHMT activity (Fig. 5A). Not surprisingly, HMECs treated with 100 μM DMF presented higher serine m + 3 labeling from labeled serine ($p < 0.0001$) (Fig. 5B). Glycine m + 2, which comes from this extracellular labeled serine, was also higher after DMF treatment ($p < 0.05$) (Fig. 5C). Nevertheless, the contribution of labeled glycine to the intracellular glycine pool was not as high in DMF-treated cells respect to the control condition compared to serine uptake ($p < 0.05$) (Fig. 5D). No significant differences were found in serine labeling from glycine (Fig. 5E). These data indicate that uptake of these two amino acids is increased in DMF-treated HMECs. Supplementary Data for these experiments are available in Supplementary Data 4.

Regarding intracellular serine and glycine pools, when serine, but not glycine, was present in the medium, intracellular serine levels were higher in DMF-treated cells ($p < 0.01$), but glycine levels were lower ($p < 0.05$) (Fig. 5F), indicating an increase in extracellular serine uptake, which could not be converted to glycine. However, depleting serine from the medium while extracellular glycine is available diminished serine levels after DMF treatment, yet not significantly, whereas the glycine pool was unexpectedly decreased ($p < 0.05$) (Fig. 5G). Supplementary Data for these experiments are available in Supplementary Data 1 and 5.

**DMF down-regulates PHGDH activity without affecting protein levels.** So far, our data indicated a major role of serine and, specially, glycine synthesis in HMECs compared to their extracellular uptake, pointing to DMF as an inhibitor of this biosynthetic pathway. This was surprising, since our proteomics analysis revealed a 4.5-fold expression of PSPH, the third and last enzyme in the synthesis pathway ($p < 0.0001$) (Supplementary Table S2). However, the rate-limiting enzyme under cell culture conditions is PHGDH, which catalyzes the committed step in the serine synthesis pathway[22]. Based on proteomics results, PHGDH protein expression was not significantly changed in DMF-treated ECs (Supplementary Table S1) and we further validated this data by Western blot, assessing that PHGDH protein expression was unaffected by DMF treatment in HMECs (Fig. 6A, B and Supplementary Fig. S12). Nonetheless, treatment with 100 μM DMF decreased PHGDH activity in HMECs ($p < 0.0001$) (Fig. 6C), and this partial inhibition matches the percentage of serine m + 3 labeling from glucose (Fig. 3D). Therefore, even if PSPH protein levels are higher with DMF, a lower PHGDH activity is most likely limiting the serine and glycine synthesis rate in HMECs. Interestingly, DMF failed to decrease PHGDH activity in cell lysates from the control condition ($1.30 ± 0.13$ mU/mg *vs.* $1.32 ± 0.13$ mU/mg in cell lysates from control cells treated *on site* with DMSO or 100 μM DMF, respectively). These data suggest that the PHGDH inhibition exerted by DMF in ECs was not a

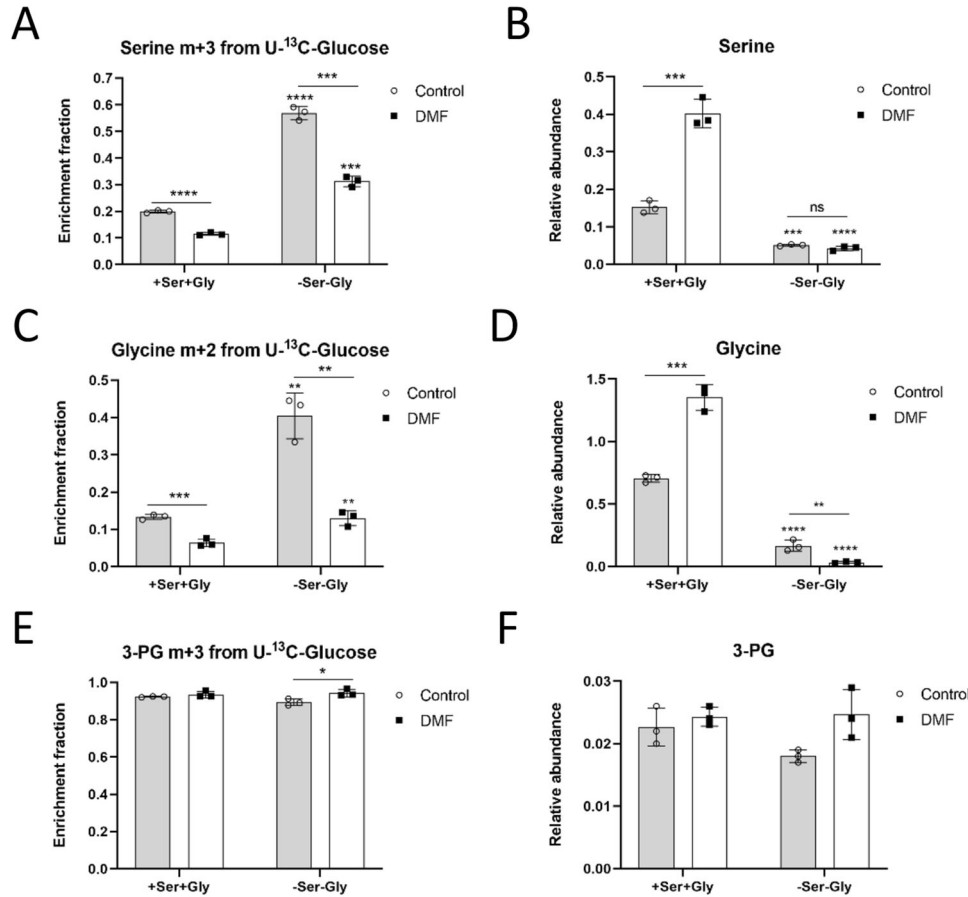

**Fig. 4 DMF favors serine and glycine synthesis from glucose in the absence of extracellular serine and glycine in HMECs. A** Fractional labeling of serine from [U-$^{13}$C]-glucose, (**B**) intracellular serine levels, (**C**) fractional labeling of glycine from [U-$^{13}$C]-glucose, (**D**) intracellular glycine levels, (**E**) fractional labeling of 3-PG from [U-$^{13}$C]-glucose and (**F**) intracellular 3-PG levels in cells treated with 100 μM DMF for 24 h in the presence or absence of extracellular serine and glycine. Data are expressed as means ± SD for an experiment with three replicates. Statistical significance was determined using the unpaired, two-sided Student $t$ test and the resulting $p$ values are shown in the figure as follows: *$p < 0.05$; **$p < 0.01$; ***$p < 0.001$; ***$p < 0.0001$ versus condition with extracellular serine and glycine. ns: non-significant.

direct effect on the enzyme, and was probably dependent on some cellular processes. Supplementary Data for these experiments are available in Supplementary Data 6–8.

**DMF alters the metabolic profile of endothelial proliferative cells in the caudal fin of zebrafish**. We used the zebrafish model for performing an in vivo experiment in order to check whether the effects seen in human ECs in vitro were also exerted in vivo. For that aim, we selected the caudal fin regeneration assay in zebrafish as detailed in the Methods section and represented in Fig. 7A. We stablished an incubation time of 24 h with non-lethal doses of DMF to mimic the conditions to which cells in culture were exposed. First, we stained the border of the amputated fins after incubation with or without DMF using Picrosirius red. The histological preparations showed that the regenerating regions of the caudal fin are enriched in proliferative blood vessels (Supplementary Fig. S10). Therefore, we assumed this experimental strategy as a valid approach for the study of proliferative microvascular ECs in this animal model.

Then, we pooled sections for the non-regenerated or regenerated caudal fin of control or DMF-treated zebrafish and analyzed their metabolite profile using $^1$H HR-MAS NMR. The spectra of relevant metabolites are shown for each experimental condition in Fig. 7B. No major changes were found in the relative abundance of either glucose and glutamine between non-regenerated and control regenerated samples, whereas a small

difference was detected for lactate, serine and glycine. Regarding treatment with DMF, this compound increased glucose, lactate, glutamine and glycine abundances at 25 and/or 50 μM, whereas no change was seen for serine (Fig. 7C). These data are consistent, with the exception of glutamine, with the results obtained in vitro. Changes greater than a 25% in comparison with the non-regenerated condition were found, as represented in Supplementary Fig. S11. Supplementary Data for these experiments are available in Supplementary Data 1.

**Discussion**
Study of EC metabolism has gained importance in the search of new targets for the treatment of angiogenesis-dependent diseases, in which microvascular ECs play an important role[8,9]. As far as we are concerned, this is the first work demonstrating the reduction on PHGDH activity by the anti-angiogenic compound DMF. The serine synthesis pathway has been described to be essential for the progression of several types of tumors, and the development of PHGDH inhibitors has emerged as a promising cancer therapy[23]. Remarkably, during the revision of this work a study showed that PHGDH-mediated serine synthesis is altered in tumor-associated ECs[24]. Other studies have reported a transcriptional regulation of PHGDH by other molecules also affected by DMF[25,26]. Nevertheless, our results suggest that the effects of DMF on microvascular EC energetic metabolism are not likely controlled by a transcriptional regulation but by direct enzymatic

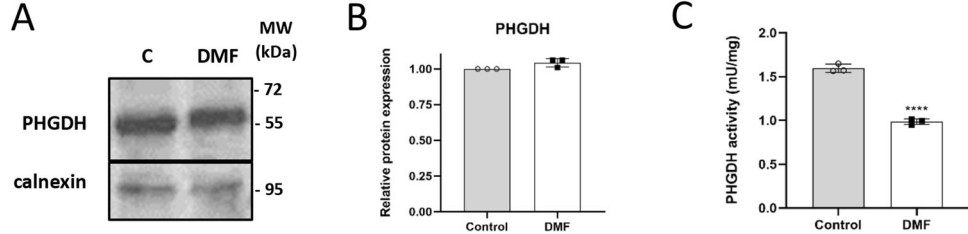

**Fig. 5 DMF favors extracellular serine and glycine uptake in HMECs. A** Scheme of serine and glycine interconversion illustrating labeling from [U-13C]-serine and [U-13C]-glycine. **B** Fractional labeling of serine and (**C**) glycine from [U-13C]-serine, and of (**D**) glycine and (**E**) from [U-13C]-glycine in cells treated with 100 μM DMF for 24 h. **F** Intracellular serine and glycine levels in medium without glycine or (**G**) without serine in cells treated with 100 μM for 24 h. Data are expressed as means ± SD for an experiment with three replicates. Statistical significance was determined using the unpaired, two-sided Student $t$ test and the resulting $p$ values are shown in the figure as follows: *$p < 0.05$; **$p < 0.01$; ***$p < 0.0001$ versus untreated control.

**Fig. 6 DMF inhibits PHGDH activity in HMECs. A** Representative Western blot for PHGDH and (**B**) quantification in cells treated with 100 μM DMF for 24 h. **C** PHGDH activity in cells treated with 100 μM for 24 h. Data are expressed as means ± SD for three independent experiments. Statistical significance was determined using the unpaired, two-sided Student $t$ test and the resulting $p$ values are shown in the figure as follows: ****$p < 0.0001$ versus untreated control.

inhibition by an unknown mechanism which is likely to require some cellular process.

DMF is known to be an electrophilic molecule able to bind to protein cysteine residues in a process called succination, hence modifying their activity[27]. Indeed, many of the DMF molecular targets suffer cysteine succination[1]. This fact makes studying the exact mechanism of action of this compound a big challenge due to this lack of specificity, which could affect many molecules through a cascade of different dysfunctional signaling pathways. Interestingly, a global analysis of cysteine ligandability performed in cancer cells revealed that Cys369 of PHGDH can react with

electrophilic small molecules[28]. Whether the modulation of PHGDH activity exerted by DMF involves succination of cysteine residues of this protein remains unstudied. However, since DMF failed to inhibit PHGDH activity in control cell lysates in vitro, it is likely that some cellular process participates in the regulation of PHGDH activity mediated by DMF and that cysteine succination might not be enough for repressing the PHGDH activity in microvascular ECs.

Our data point out that suppression of PHGDH activity in microvascular ECs seems to have a greater effect on glycine synthesis compared to serine synthesis, since glycine levels were

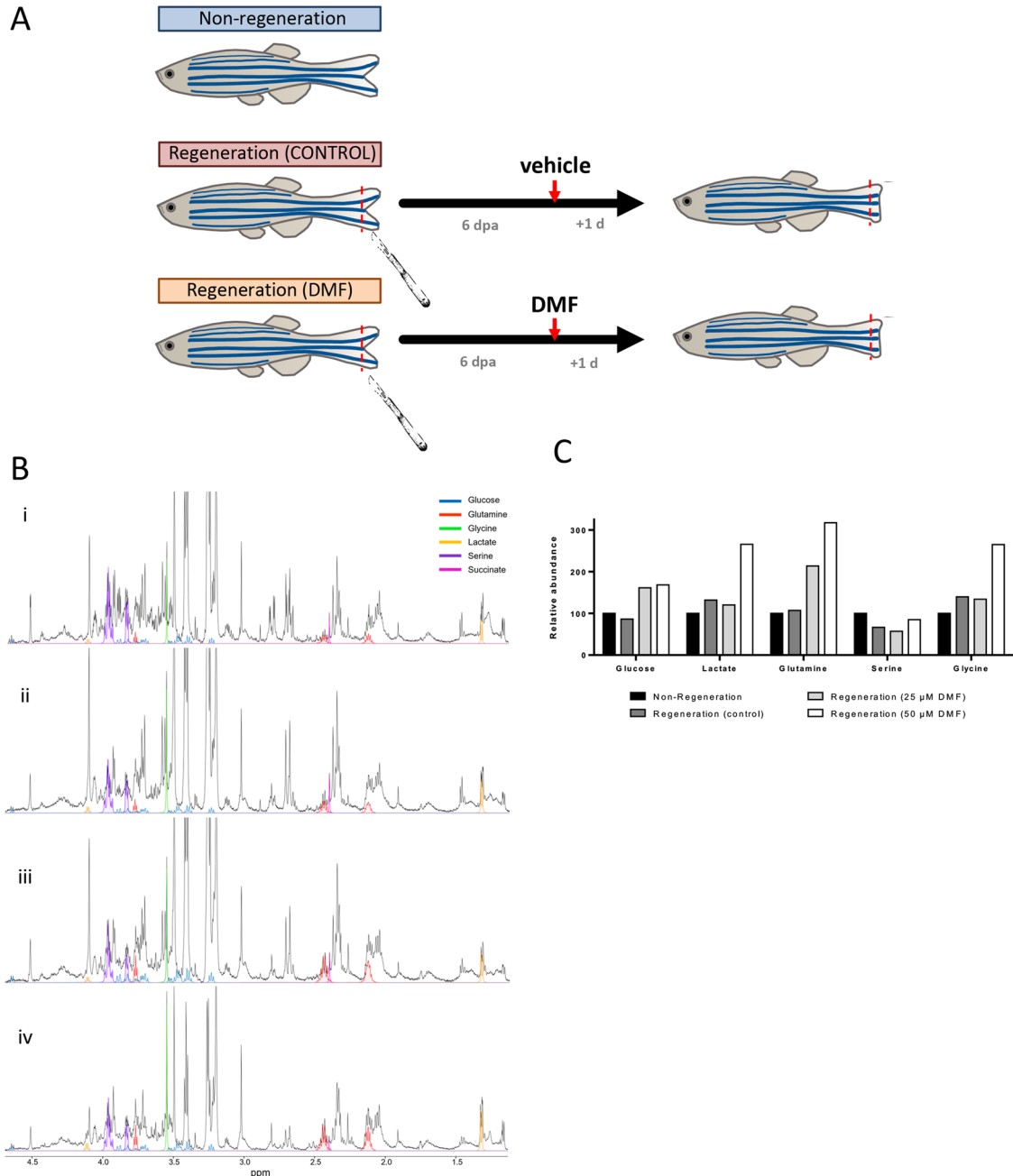

**Fig. 7 DMF alters metabolism of endothelial cells from caudal fin of zebrafish. A** Schematics for the experimental conditions used in the fin regeneration assay. **B** Quantitative analysis of [1]H HR-MAS NMR spectra from intact zebrafish fin using Chenomx software. The four experimental conditions shown in the figure are (i) non-regeneration, (ii) regeneration condition treated with the vehicle, (iii) and (iv) regeneration condition treated with 25 µM and 50 µM of DMF, respectively. For the sake of clarity, only relevant metabolite fittings are shown: glucose (blue), glutamine (red), glycine (green), lactate (yellow), serine (purple) and succinate (pink). **C** Relative abundance compared with the non-regenerated condition for glucose, lactate, glutamine, serine and glycine metabolite. Data are the sum of a pool of sections of the caudal fin from 20 specimens, independently of their biological sex. The amount of material for each experimental condition was as follows: 15.2 mg for non-regenerated fish, 17.6 mg for vehicle condition, 17.8 mg for 25 µM DMF and 15.2 mg for 50 µM DMF.

more compromised after DMF treatment than the serine pool even when extracellular glycine was available. It is known that de novo mitochondrial glycine synthesis is highly active in ECs and more important than cellular uptake[29]. Conversely, extracellular glycine has been shown to stimulate VEGF signaling and angiogenesis in vitro and in vivo by promoting mitochondrial function, and VEGF was found to promote the expression of the glycine transporter GlyT1 in ECs, while it did not affect the levels of the

enzymes involved in glycine synthesis[30]. DMF was reported ten years ago to exert an anti-angiogenic activity in vitro and in vivo, at least partially due to VEGFR2 suppression[4,5]. The exact mechanism by which DMF represses VEGFR2 expression remains unexplored and would require further research. Since DMF inhibits VEGFR2 and, therefore, the VEGF signaling pathway in ECs, it could be possible that glycine uptake is suppressed in DMF-treated ECs. On top of that, both extracellular

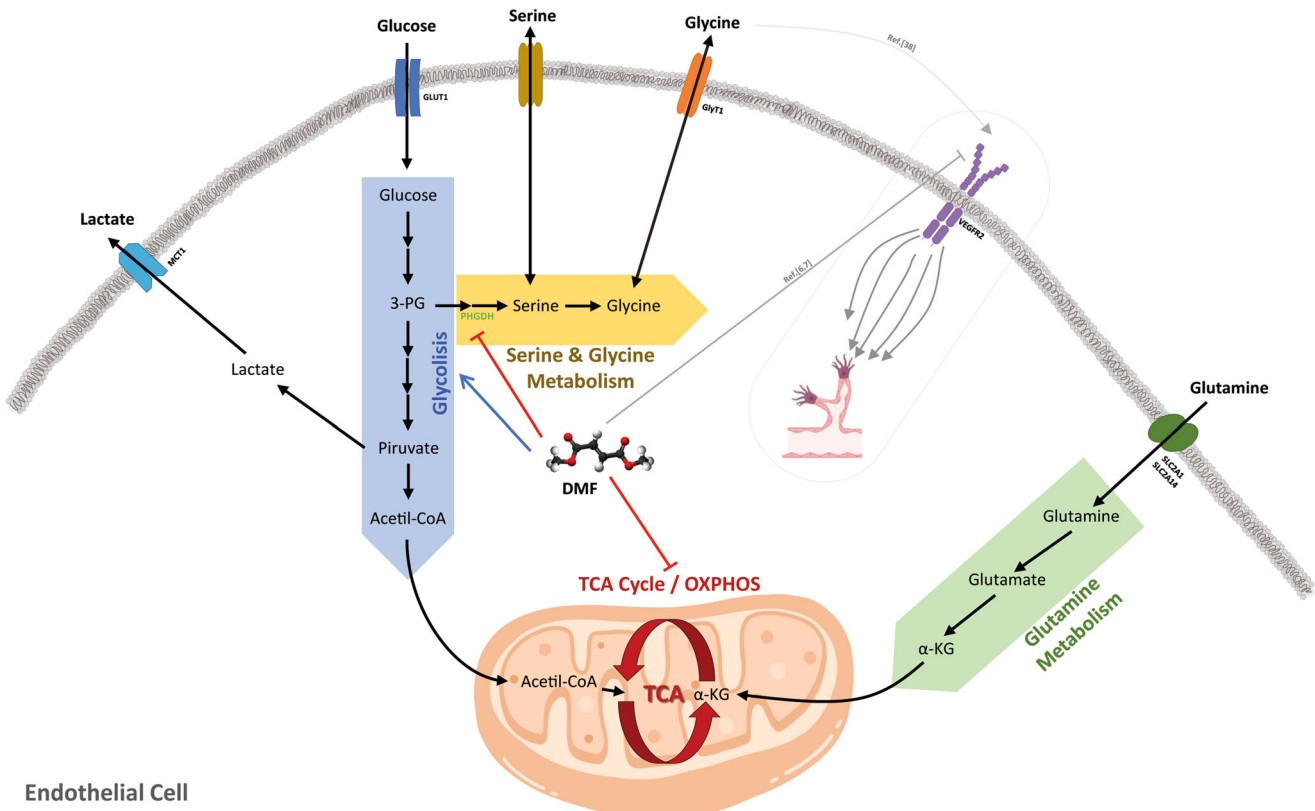

**Fig. 8 Summary of DMF effects on energetic EC metabolism.** The results of this work show how DMF induces glucose uptake through upregulation of GLUT1 expression. This increase in glucose uptake leads to a greater glycolytic rate and lactate secretion to the extracellular media. Conversely, glutamine uptake is diminished in cells treated with DMF, and its oxidation through the TCA cycle is reduced. DMF diminished PHGDH activity and synthesis of serine and glycine from glucose. It is described in the bibliography that DMF inhibits VEGFR2 expression, and that extracellular glycine has been seen to promote VEGF signaling and angiogenesis through upregulation of mitochondrial activity. Whether inhibition of glycine metabolism is related with the anti-angiogenic activity of DMF remains unclear.

glycine and de novo synthetized glycine seem important to EC metabolism and function, and DMF plays a major role in glycine metabolism through PHGDH inhibition in these cells. Nonetheless, the existence of an interplay between the DMF-induced PHGDH inhibition and the anti-angiogenic activity of this molecule remains unclear.

Among other metabolic pathways, glycolysis has been described to be essential for vessel sprouting[31]. Some years ago, DMF was found to inhibit glycolysis in immune cells through targeting of GAPDH activity mediated by succination of several cysteine residues[12]. Conversely, the results found in this work show that DMF increases glycolysis whereas diminishes OXPHOS in microvascular ECs. Why this compound affects energetic metabolism differently in different cell types has yet to be answered. Remarkably, using ECs, Vandekeere and colleagues found out that silencing PHGDH impaired angiogenesis, even when the glycolytic rate of PHGDH knock-down cells was higher than those whose levels of PHGDH remained intact[32]. These results are similar to those obtained in this work for the inhibition of PHGDH after DMF treatment in HMECs. Thus, it is likely that PHGDH inhibition boosts glycolytic activity in order to compensate the reduction in serine synthesis rate. However, we found an increased glucose uptake in several cell lines treated with DMF, including MDA-MB-231, a triple negative breast cancer cell line which lacks PHGDH[33]. Therefore, additional mechanisms must regulate the increase in the glycolytic rate exerted by DMF. In addition, although glycolysis has been shown to be essential for angiogenesis, inhibition of PHGDH impaired the

angiogenic process but increased the glycolytic rate in ECs[31,32]. The results published by Vandekeere et al. and the data presented in this work reinforces the complexity of metabolic regulation in ECs and its relation with the angiogenic switch.

It is worthy of note the different effects of DMF on glutamine oxidation in different cell lines. DMF diminished glutamine oxidation in different EC lines (including microvascular and mesovascular ECs) and in cancer MDA-MB-231 cells. All these cell lines are known to be highly glycolytic[34–36]. However, DMF failed to alter glutamine oxidation in the highly glutamine-dependent, oxidative cell line HeLa[37]. This differential effect points out to a different regulation of energetic metabolism depending on the metabolic preferences of the cells.

We also tested the role of DMF in vivo. For that aim, we performed a fin regeneration assay using zebrafish as an animal model. DMF was administered in the fish water, so that the fins were in direct contact with this compound, a common practice when this animal model is used[38,39]. This route of administration differs from that given to humans, where orally ingested DMF is rapidly metabolized to monomethyl fumarate (MMF) in the gastrointestinal tract[40]. This limitation should always be taken into account, since this in vivo model might not fully replicate DMF exposure in the clinical. Nevertheless, we consider that our data present sufficient physiologic relevance for several reasons. First of all, the levels of some metabolites after fin amputation and DMF administration remained unaltered and similar to the non-regeneration, control condition, as it can be seen in Supplementary Fig. S11 (for example, choline or ascorbate). If the changes observed

after DMF treatment were not physiologic, a more or less major change in the levels of all metabolites could have been expected. Moreover, several of those changes are dose-dependent. These facts demonstrate that the changes produced by DMF are, at some extent, specific. Accordingly, some metabolites such as glucose, lactate and glycine followed the same pattern as HMECs in vitro.

All in all, the results obtained in this work make an interesting point in using DMF as a therapeutic tool in different pathological contexts, since this compound may exert different effects in different cell types such as ECs, cancer cells and immune cells. We consider important to remark that these effects may be also different in different kind of ECs, since their metabolic profile may change depending not only on their proliferative capacity, but also on their tissue localization[41]. Therefore, this work opens a new field of research on the effect of DMF on EC metabolism and its possible but not studied effect on the angiogenic switch.

## Conclusions

Altogether, our data suggest a complex regulation of microvascular EC metabolism by DMF (Fig. 8). The molecular mechanism by which DMF inhibits PHDGH, and whether this inhibition affects tube formation, remains uncertain, but the effect on serine and glycine synthesis in microvascular ECs is solid. Elucidating the exact mechanism of action of DMF requires a vast and comprehensive experimental design, due to its reported high molecular reactivity and wide regulation of transcription factors, such as Nrf2 and NF-κB. In conclusion, the results presented in this work point to a metabolic regulation of DMF in microvascular ECs, which could affect to their function.

## Data availability

Our raw proteomic data are accessible in ProteomeXchange with the accession number PXD045447. Metabolomic raw data is provided in the form of 8 Excel data files in Supplementary Material (Supplementary Data 1–8). Uncropped immunoblot images from Fig. 6 panel A and Supplementary Fig. S5 are available as Supplementary Fig. S12 in the pdf file "Supplementary information".

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

## Acknowledgements

We thank Maria Somoza for her assistance with NMR experiments. NMR experiments were performed in the ICTS "NANBIOSIS" (U28 Unit at the Andalusian Center for Nanomedicine & Biotechnology, BIONAND). M.O. was recipient of a predoctoral FPU grant from the Spanish Ministry of Education, Culture and Sport. M.B. was supported by Juan de la Cierva – Incorporation Program (IJC2018-037657-I), Spanish Ministry of Science and Innovation. C. Caro was supported by a senior postdoctoral grant from the Andalusian Government (RH-0040-2021). This work was supported by grants PID2019-105010RB-I00 (MICINN and FEDER), UMA18-FEDERJA-220, PY20_00257 (Andalusian Government and FEDER), PID2020-118448RBC21 (MICN and AEI), P20_00727/PAIDI2020 (Andalusian Government) and funds from group BIO 267 (Andalusian Government). Currently, our group has no grant funding our research. The "CIBER de Enfermedades Raras" and "CIBER de Enfermedades Cardiovasculares" are an initiative from the ISCIII (Spain). R.J.D. is funded by Howard Hughes Medical Institute and the National Cancer Institute (Grant R35CA22044901). The funders had no role in the study design, data collection and analysis, decision to publish or preparation of the manuscript.

## Author contributions

M.O., C.Y., B.M-P.; R.J.D., A.R.Q. and M.A.M. designed the research; M.O. and C.Y. performed the experiments and analyzed the data; H.S.V. performed LC/MS labeling and metabolomics assays; C. Cárdenas performed proteomics assays and analysis; M.B, C. Caro, A.D and M.L.M-G designed, performed and analyzed all experiments involving zebrafish. M.O. wrote the original draft; M.B., B.M-P., R.J.D., A.R.Q. and M.A.M. reviewed and edited the final version of the manuscript; all authors reviewed the results and approved the final version of the manuscript.

## Competing interests

The authors declare no competing interests.

## Additional information

**Peer review information** : *Communications Biology* thanks the anonymous reviewers for their contribution to the peer review of this work. Primary Handling Editors: Gabriela da Silva Xavier and Manuel Breuer. A peer review file is available.

