## [Peer Review File · Communications Biology]

Reviewers' comments:

Reviewer #1 (Remarks to the Author):

This manuscript addresses the metabolic consequences of DMF treatment in human microvascular endothelial cells (HMECs), reporting a number of metabolic derangements. Among these, DMF (at 50 or 100 μ M) increased glycolytic flux and decreased oxidative phosphorylation, which was associated with increased glucose uptake and decreased glutamine oxidation. Changes were also observed with regard to serine and glycine metabolism, with DMF inducing higher intracellular pools of these amino acids via increased uptake, while decreasing serine/glycine synthesis from glucose and leading to decreased PHGDH activity.

Overall, data regarding the reported metabolic changes are convincing, and understanding the metabolic consequences of DMF treatment in distinct cell types is clinically relevant and of broad interest. I have two primary critiques and some minor concerns about data presentation:

1. The biggest concern in interpreting the data is uncertain physiologic relevance. All experiments were performed with either 50 or 100 μ M DMF, but it is almost certain that endothelial cells are never exposed to such high concentrations after oral administration. After oral administration of DMF, no DMF can be measured in the circulation – only monomethyl fumarate (MMF) is found in blood (likely due to hydrolysis from GI esterases), with a maximum concentration of approximately 30 μ M (reviewed in Linker and Haghikia 2016, PMID 27433310). Although some DMF may make it to tissues (as evidenced by DMF cysteine adducts found in urine and in some tissues in animal studies), these concentrations are likely minimal. In the in vitro studies presented here, MMF was not examined, and DMF is known to be much more electrophilic with a broader array of targets compared to MMF. Without corroborating in vivo studies evaluating EC metabolism after oral DMF administration, the physiologic relevance of the findings reported here (using excessive doses of DMF in vitro) is very uncertain.
2. Although the reported metabolic changes are convincing, the data are descriptive, and no evidence is presented to determine the causal relation between the observed metabolic changes and the functional outcome of interest – namely, angiogenesis and tube formation. Therefore, the functional relevance of the observed metabolic changes remains uncertain.

Minor critiques:

Rigor and transparency:

-Although it's understandable why representative data must be shown for certain experiments, such as the extracellular flux assays, confidence in the rigor of the results would be increased by also showing cumulative data in addition to the single representative experiment, even if that requires normalization of data within each experiment. At the very least, the Seahorse curves from the other experiments should be included as supplementary data if no cumulative data is presented. Similarly, it appears that results from the NOVA/ammonia assays are shown only from a single experiment, although the methods section states the experiment was "repeated several times." The number of times it was repeated should be explicitly stated, and cumulative data should be shown if at all possible.

-The number of independent experiments from which data were derived should be explicitly stated for each set of data, rather than making a blanket statement in the methods that experiments were repeated "at least three" times.

Reviewer #2 (Remarks to the Author):

Dimethyl fumarate (DMF), an approved treatment for relapsing-remitting multiple sclerosis and psoriasis, exerts pleiotropic effects on various immune cells and endothelial cells (EC). In psoriasis, DMF inhibits the expression of adhesion molecules, resulting in impaired rolling and adhesion of lymphocytes and thereby preventing the critical step of lymphocyte recruitment to sites of inflammation. In a previous publication, the authors demonstrated that DMF reduces the formation of capillary-like structures in VEGF-stimulated human endothelial cells and inhibits angiogenesis (vessel formation) both in vitro and in vivo (García-Caballero et al. 2011). Given the importance of EC metabolism in angiogenesis, the authors examined in this publication the effect of DMF on the energy metabolism of human microvascular ECs (HMEC). Here, they show that DMF induces a metabolic switch towards a glycolytic phenotype, whereas mitochondrial respiration was reduced in HMECs. The reduction of mitochondrial respiration is linked to decreased serine and glycine synthesis through inhibition of PHGDH activity.

The study by Ocaña et al. describes an unknown mechanism of action of DMF in HMEC. Overall, the results are of great interest to the research field. However, there are also some concerns.

Major points:

- The described metabolic changes in EC by DMF were observed exclusively in vitro, but they should be confirmed in an appropriate in vivo model.
- In their previous paper, the authors showed that high DMF concentrations lead to increased apoptosis of BAECs and HUVECs. Since DMF can also induce apoptosis in a variety of other cells, the authors should clearly demonstrate that this is not the reason for the observed metabolic changes in HMECs.
- A decreased amount of serine/glycine leads to reduced intracellular GSH levels. DMF is known to bind GSH directly. So to what extent does this observation fit with the data described in this paper? Furthermore, the enzyme that produces GSH, GCLC, is increased 7-fold in DMF-treated HMECs. Please investigate this aspect in more detail.
- Several studies could show that increased expression of HMOX1 in HUVECs is induced mainly by oxidative stress. Thus, could the 26-fold upregulation of HMOX1 in HMECs after DMF addition described here also be the consequence of mitochondrial dysfunction. Please investigate this connection.
- Especially the HMECs seem to have a reduced glutamine metabolism under DMF, whereas the other endothelial cells such as HUVECs are not affected. What could be the reason for this, because as described in the previous paper from 2011, DMF does seem to inhibit angiogenesis in both cells.

Minor points:

- Please indicate the statistical test used in each figure legend.
- Since ECs have very low metabolic activity, which is difficult to detect in the Seahorse Analyzer, please explain the method of normalization used and show the original data of individual measurements as examples.
- Please change your figures according to the submission guidelines and show either individual data points overlaid on the bars or convert the bar chart to a dot-plot or box-and-whisker-plot format.

Reviewer #1

This manuscript addresses the metabolic consequences of DMF treatment in human microvascular endothelial cells (HMECs), reporting a number of metabolic derangements. Among these, DMF (at 50 or 100 μ M) increased glycolytic flux and decreased oxidative phosphorylation, which was associated with increased glucose uptake and decreased glutamine oxidation. Changes were also observed with regard to serine and glycine metabolism, with DMF inducing higher intracellular pools of these amino acids via increased uptake, while decreasing serine/glycine synthesis from glucose and leading to decreased PHGDH activity.

Overall, data regarding the reported metabolic changes are convincing, and understanding the metabolic consequences of DMF treatment in distinct cell types is clinically relevant and of broad interest. I have two primary critiques and some minor concerns about data presentation.

Major critiques

1. The biggest concern in interpreting the data is uncertain physiologic relevance. All experiments were performed with either 50 or 100 μ M DMF, but it is almost certain that endothelial cells are never exposed to such high concentrations after oral administration. After oral administration of DMF, no DMF can be measured in the circulation – only monomethyl fumarate (MMF) is found in blood (likely due to hydrolysis from GI esterases), with a maximum concentration of approximately 30 μ M (reviewed in Linker and Haghikia 2016, PMID 27433310). Although some DMF may make it to tissues (as evidenced by DMF cysteine adducts found in urine and in some tissues in animal studies), these concentrations are likely minimal. In the *in vitro* studies presented here, MMF was not examined, and DMF is known to be much more electrophilic with a broader array of targets compared to MMF. Without corroborating *in vivo* studies evaluating EC metabolism after oral DMF administration, the physiologic relevance of the findings reported here (using excessive doses of DMF *in vitro*) is very uncertain.

We really appreciate Reviewer 1 concern about the lack of an *in vivo* model in our work. Therefore, we have designed an animal model using zebrafish. Specifically, we adapted the caudal fin regeneration assay in order to analyze the metabolites present in the cells of the regenerating border, enriched in proliferative endothelial cells due to new blood vessel formation, after treatment with DMF. Please find our results in Figures 7, S10 and S11 of the amended version of the manuscript.

2. Although the reported metabolic changes are convincing, the data are descriptive, and no evidence is presented to determine the causal relation between the observed metabolic changes and the functional outcome of interest – namely, angiogenesis and tube formation. Therefore, the functional relevance of the observed metabolic changes remains uncertain.

We appreciate this major point made by Reviewer 1. Regrettably, the scope of our manuscript is not to determine the causal relation between the effects of DMF in EC metabolism and its anti-angiogenic role. In fact, we do not provide data to establish whether that relation actually exists. In this work, our goal was to discover whether this well-studied and highly reactive anti-angiogenic compound was also able to alter at some extent EC metabolism. For that purpose, we first wanted to be sure that DMF was able to inhibit tube formation in the microvascular EC model we used for the metabolic experiments, and we included these results as part of the

supplementary data of the manuscript, but not in the main figures, indicating that the anti-angiogenic activity of DMF was not a central aim. We consider that the data we present in this manuscript successfully answer to the question we wanted to answer (alteration of EC metabolism by DMF). Our data reveals that DMF indeed disturbs EC metabolism through changes in central metabolic pathways, such as glycolysis and glutamine metabolism, and in other minor, yet not less important, routes, namely serine and glycine *de novo* synthesis. As we point in our manuscript, these changes may or not be directly related with the anti-angiogenic activity of DMF and further research may be necessary to elucidate the precise mechanism of action of this drug. We hope that the data we provide help other researchers to deepen the direct targets of DMF activity.

Minor critiques:

Rigor and transparency:

-Although it's understandable why representative data must be shown for certain experiments, such as the extracellular flux assays, confidence in the rigor of the results would be increased by also showing cumulative data in addition to the single representative experiment, even if that requires normalization of data within each experiment. At the very least, the Seahorse curves from the other experiments should be included as supplementary data if no cumulative data is presented. Similarly, it appears that results from the NOVA/ammonia assays are shown only from a single experiment, although the methods section states the experiment was "repeated several times." The number of times it was repeated should be explicitly stated, and cumulative data should be shown if at all possible.

We sincerely appreciate this comment from Reviewer 1. We completely understand the concern for transparency, and hence we have included as supplemental figures the remaining experiments for Seahorse and NOVA/ammonia measurements. Please find also all the corresponding figures below in this letter. Additionally, we have included in each figure legend of the amended version of the manuscript the exact number of experiments that were performed for each data set.

Figure 1A of the amended version of the manuscript.

Figures S2A and S2B of the amended version of the manuscript.

Figure 1B of the amended version of the manuscript.

Figures S2C and S2D of the amended version of the manuscript.

Figure 1D of the amended version of the manuscript.

Figures S1E and S1F of the amended version of the manuscript.

Figure 1E of the amended version of the manuscript.

Figures S1G and S1H of the amended version of the manuscript.

Figure 1F of the amended version of the manuscript.

Figures S1I and SIJ of the amended version of the manuscript.

Figure S3B of the amended version of the manuscript.

Figure S3D of the amended version of the manuscript.

-The number of independent experiments from which data were derived should be explicitly stated for each set of data, rather than making a blanket statement in the methods that experiments were repeated “at least three” times.

As said in the last point, in the amended version of the manuscript we have included the number of independent experiments and replicates performed for each set of data in each figure legend.

Reviewer #2

Dimethyl fumarate (DMF), an approved treatment for relapsing-remitting multiple sclerosis and psoriasis, exerts pleiotropic effects on various immune cells and endothelial cells (EC). In psoriasis, DMF inhibits the expression of adhesion molecules, resulting in impaired rolling and adhesion of lymphocytes and thereby preventing the critical step of lymphocyte recruitment to sites of inflammation. In a previous publication, the authors demonstrated that DMF reduces the formation of capillary-like structures in VEGF-stimulated human endothelial cells and inhibits angiogenesis (vessel formation) both *in vitro* and *in vivo* (García-Caballero et al. 2011). Given the importance of EC metabolism in angiogenesis, the authors examined in this publication the effect of DMF on the energy metabolism of human microvascular ECs (HMEC). Here, they show that DMF induces a metabolic switch towards a glycolytic phenotype, whereas mitochondrial respiration was reduced in HMECs. The reduction of mitochondrial respiration is linked to decreased serine and glycine synthesis through inhibition of PHGDH activity.

The study by Ocaña et al. describes an unknown mechanism of action of DMF in HMEC. Overall, the results are of great interest to the research field. However, there are also some concerns.

Major points:

- **The described metabolic changes in EC by DMF were observed exclusively *in vitro*, but they should be confirmed in an appropriate *in vivo* model.**

We really appreciate Reviewer 2 concern about the lack of an *in vivo* model in our work. Therefore, we have designed an animal model using zebrafish. Specifically, we adapted the caudal fin regeneration assay in order to analyze the metabolites present in the cells of the regenerating border, enriched in proliferative endothelial cells due to new blood vessel formation. Please find our results in Figures 7, S10 and S11 of the amended version of the manuscript.

- **In their previous paper, the authors showed that high DMF concentrations lead to increased apoptosis of BAECs and HUVECs. Since DMF can also induce apoptosis in a variety of other cells, the authors should clearly demonstrate that this is not the reason for the observed metabolic changes in HMECs.**

This point made by Reviewer 2 is quite appropriate and we totally understand this concern. Previously to test the effects of DMF on EC metabolism, we performed cell cycle experiments using the maximum dose of DMF (100 μ M) during 24 h in HMECs. The results showed that 100 μ M did not statistically induce apoptosis in these cells (p -value = 0,14).

Figure. HMECs were exposed for 24 h to 100 μ M DMF, stained with propidium iodide and percentages of cells on subG1, G0/G1 and S/G2/M phases were determined using a FACS VERSE™ cytometer. Data are expressed as means \pm SD of three independent experiments.

We hope Reviewer 2 consider these results as indicative of the non-toxic activity of DMF in our cell model.

• A decreased amount of serine/glycine leads to reduced intracellular GSH levels. DMF is known to bind GSH directly. So to what extent does this observation fit with the data described in this paper? Furthermore, the enzyme that produces GSH, GCLC, is increased 7-fold in DMF-treated HMECs. Please investigate this aspect in more detail.

We really appreciate this comment from Reviewer 2. This point is a very interesting matter, and we have additional experiments about the effect of DMF on serine and glycine metabolism and its connection with GSH synthesis. However, those experiments are part of a non-published manuscript along with other results, which we hope to be published soon.

• Several studies could show that increased expression of HMOX1 in HUVECs is induced mainly by oxidative stress. Thus, could the 26-fold upregulation of HMOX1 in HMECs after DMF addition described here also be the consequence of mitochondrial dysfunction. Please investigate this connection.

DMF is known to be a highly thiol-reactive electrophile molecule. As has been previously reviewed (Saidu et al. 2019, doi:10.1002/med.21567), one of the targets of DMF is cysteine 151 of the KEAP1 protein. This interaction induces conformational alterations of KEAP1 protein, which causes a disruption of the KEAP1-NRF2 interaction, and hence allowing the translocation of the transcription factor NRF2 into the nucleus. Once inside the nucleus, NRF2, through binding to the antioxidant response element (ARE), is able to induce the activity of several enzymes, including heme oxygenase-1 (HMOX-1).

Therefore, some of the changes in gene or protein expression exerted by DMF may be the result of its electrophilic activity and not necessarily a consequence of its effects on EC metabolism. Indeed, we consider of great interest the most-likely consequences of this alteration of gene/protein expression to EC metabolism, rather than the other way around. However, this point lies outside the goals of our manuscript, but we hope we open new questions for other researchers to look into it.

- **Especially the HMBECs seem to have a reduced glutamine metabolism under DMF, whereas the other endothelial cells such as HUVECs are not affected. What could be the reason for this, because as described in the previous paper from 2011, DMF does seem to inhibit angiogenesis in both cells.**

We appreciate this comment from Reviewer 2. We have revised our raw data from glutamine oxidation in HUVECs. We repeated the experiments three times, and the data we obtained, which is represented in Supplementary Figure S3 of the previous version of this manuscript, are the following:

0.0068 % glutamine oxidized in control conditions versus 0,0078 % with 100 μ M DMF

0.0129 % glutamine oxidized in control conditions versus 0,0045 % with 100 μ M DMF

0.0090 % glutamine oxidized in control conditions versus 0,0031 % with 100 μ M DMF

The mean values are 0.0096 % for control cells versus 0.0051 % for DMF-treated cells. Therefore, DMF actually halves glutamine oxidation in HUVECs (53% of glutamine oxidation compared to the control condition). However, we cannot say that this change is statistically significant due to a deviation of one of the replicates. This may be due to the instability of primary cell cultures compared to immortalized cell lines.

In order to maintain transparency, we also provide here the data of the three independent replicates using HMECs, which are represented in Figure 1G of the previous version of this manuscript:

0.20 % glutamine oxidized in control conditions versus 0,12 % with 100 μ M DMF

0.24 % glutamine oxidized in control conditions versus 0,13 % with 100 μ M DMF

0.19 % glutamine oxidized in control conditions versus 0,11 % with 100 μ M DMF

The mean values are 0.21 % for control cells versus 0.12 % for DMF-treated cells. As in HUVECs, glutamine oxidation almost halves in the presence of 100 μ M DMF in HMECs (57.1% compared to the control condition).

We hope that Reviewer 2 finds these data useful and that we have properly answered his/her question.

Minor points:

- **Please indicate the statistical test used in each figure legend.**

Now this information is given in each figure legend of the amended version of the manuscript.

- **Since ECs have very low metabolic activity, which is difficult to detect in the Seahorse Analyzer, please explain the method of normalization used and show the original data of individual measurements as examples.**

For the Seahorse experiments, we normalized the data using protein measurements. After data acquisition, extract proteins were extracted by adding 75 μ L RIPA buffer (50 mM Tris-HCl pH 7.4, 150 nM NaCl, 1% Triton X-100, 0.25% sodium deoxycholate and 1 mM EDTA) to each well on ice for 20 min, centrifuged at 10,000 g at 4° C for 5 min, supernatants were collected and protein quantification was performed using the DC Protein Assay from BioRad, based on the Lowry assay, following the manufacturer's instructions.

We show here the raw data and the data normalized to protein amount (in μg) of the data represented in Figure 1A of the previous version of this manuscript as an example:

Time (minutes)	ECAR (mpH/min)					
	Control ECAR			DMF ECAR		
1.437023	4.024254	0.025972	0.856093	1.750503	0.946594	4.318438
10.23648	2.158081	3.471926	5.169518	3.597612	0.792278	0.724365
18.99485	0.873985	1.771852	3.632702	2.949555	0.566683	-0.059162
27.84007	1.939495	1.926661	2.028224	3.275177	2.964324	2.802333
36.59636	1.835247	1.810301	1.895251	3.007783	2.891722	2.618645
45.34563	1.995936	1.749953	1.904717	2.925294	2.853486	2.481067
54.169	2.544253	2.415339	2.555975	3.519534	3.564177	3.095117
62.93465	2.561934	2.462271	2.507087	3.282285	3.330225	3.142055
71.68939	2.603591	2.527427	2.596001	0.368109	3.486758	3.237401
80.54995	3.695828	0.413937	0.453042	0.341663	4.904254	2.821154
89.30885	-0.576935	-0.566756	0.079965	-0.574713	0.392537	-0.807467
98.06956	-1.080747	-1.004297	-0.232352	-1.057146	-0.373719	-1.452571

$\mu\text{g}/\text{protein}$	
Control ECAR	DMF ECAR
4.39	4.09
4.47	4.7
4.24	4.7

Time (minutes)	ECAR (mpH/min/ μg protein)					
	Control ECAR			DMF ECAR		
1.437023	0.916687	0.581029	2.019087	0.427996	0.201403	0.918817
10.23648	0.049159	0.776717	1.219226	0.879612	0.016857	0.015412
18.99485	0.199085	0.396387	0.856769	0.721163	0.120571	-0.125877
27.84007	4.417985	4.310203	4.783548	8.007767	6.307072	5.962411
36.59636	4.180517	4.049889	4.469931	7.353992	6.152601	5.571586
45.34563	0.454655	3.914884	4.492256	7.152309	6.071247	5.278866
54.169	5.795566	5.403442	6.028243	8.605217	7.583354	6.585356
62.93465	5.835841	5.508436	0.591294	8.025147	7.085585	6.685223
71.68939	0.593073	5.654199	6.122645	9.000219	7.418635	6.888087
80.54995	0.841874	0.926034	1.068495	0.835362	1.043458	0.600246
89.30885	-0.013142	-0.126791	0.001886	-0.140517	0.083519	-0.171801
98.06956	-0.246184	-0.224675	-0.548	-0.258471	-0.079515	-0.309058

- **Please change your figures according to the submission guidelines and show either individual data points overlaid on the bars or convert the bar chart to a dot-plot or box-and-whisker-plot format.**

The figures in the amended version of the manuscript have been changed and now individual data points are shown in the bar charts along with the mean and standard deviation. For Seahorse curves, individual data points are displayed.

Reviewers' comments:

Reviewer #1 (Remarks to the Author):

The authors have addressed several critiques. The description of data and statistical analyses is much improved, which has increased transparency and confidence in the results. Their response that establishing causation is beyond the scope of the paper is reasonable. I appreciate the use of an in vivo model and the additional effort it took to produce that data. Overall the work is improved, as is confidence in the findings. My only critique is that the particular in vivo model chosen still leaves open the question of physiologic relevance to orally administered DMF, which is how DMF is delivered in humans. The zebrafish were maintained in water containing high concentrations of DMF, such that tail fins were exposed to concentrations of un-metabolized drug that are never present systemically in humans or other mammals after oral administration. It remains possible that the particular metabolic changes observed are an artifact of high DMF concentrations.

Reviewer #1

The authors have addressed several critiques. The description of data and statistical analyses is much improved, which has increased transparency and confidence in the results. Their response that establishing causation is beyond the scope of the paper is reasonable. I appreciate the use of an *in vivo* model and the additional effort it took to produce that data. Overall the work is improved, as is confidence in the findings. My only critique is that the particular *in vivo* model chosen still leaves open the question of physiologic relevance to orally administered DMF, which is how DMF is delivered in humans. The zebrafish were maintained in water containing high concentrations of DMF, such that tail fins were exposed to concentrations of un-metabolized drug that are never present systemically in humans or other mammals after oral administration. It remains possible that the particular metabolic changes observed are an artifact of high DMF concentrations.

We are sincerely glad that our efforts to improve our manuscript have been taken into account. Developing a new *in vivo* model in our lab has required a considerable amount of time until we achieved our goal to see the effect of DMF on the metabolism of proliferating endothelial cells in the caudal fin of zebrafish. We understand the concerns of Reviewer #1 regarding the physiologic relevance of these results. In this *in vivo* model we established an incubation time of 24 hours with DMF, to mimic the conditions to which cells in culture were exposed. Moreover, the DMF concentrations used for this *in vivo* experiment did not were as high as to compromise the viability of these animals but, in fact, the metabolic profile changed. Several of those changes are dose-dependent and the changes observed in our *in vivo* model are similar of those obtained in the *in vitro* assays. Moreover, the levels of some metabolites after fin amputation and DMF administration remain unaltered and similar to the non-regeneration, control condition, as it can be seen in Supplementary Figure S11 (for example, choline or ascorbate). If the changes observed after DMF treatment were not physiologic, we would have expected a more or less major change in the levels of all metabolites. The fact that some metabolites are unaltered demonstrate that the changes produced by DMF are, at some extent, specific. We hope that Reviewer #1 finds this explanation enough for considering the physiologic relevance of our results.

REVIEWERS' COMMENTS:

Reviewer #1 (Remarks to the Author):

I do understand the additional effort required to establish an in vivo model. The findings are relevant. I think the Discussion should be revised in a minor fashion to state the caveat that the in vivo model might not fully replicate DMF exposure in humans given that direct exposure to DMF appears to be low when administered orally.